# Long-Term Lithium Therapy: Side Effects and Interactions

**DOI:** 10.3390/ph16010074

**Published:** 2023-01-03

**Authors:** Ewa Ferensztajn-Rochowiak, Janusz K. Rybakowski

**Affiliations:** Department of Adult Psychiatry, Poznan University of Medical Sciences, 60-744 Poznan, Poland

**Keywords:** lithium, bipolar disorder, long-term therapy, side effects, interactions

## Abstract

Lithium remains the drug of first choice for prophylactic treatment of bipolar disorder, preventing the recurrences of manic and depressive episodes. The longitudinal experiences with lithium administration greatly exceed those with other mood stabilizers. Among the adverse side effects of lithium, renal, gastrointestinal, neurological, thyroid, metabolic, cognitive, dermatological, cardiologic, and sexual are listed. Probably, the most important negative effect of lithium, occurring mostly after 10–20 years of its administration, is interstitial nephropathy. Beneficial side-effects of long-term lithium therapy also occur such as anti-suicidal, antiviral, and anti-dementia ones. Pharmacokinetic and pharmacodynamic interactions of lithium, mostly those with other drugs, may have an impact on the success of long-term lithium treatment. This paper makes the narrative updated review of lithium-induced side-effects and interactions that may influence its prophylactic effect in bipolar disorder. Their description, mechanisms, and management strategies are provided. The papers appearing in recent years focused mainly on the long-term lithium treatment are reviewed in detail, including recent research performed at Department of Psychiatry, Poznan University of Medical Sciences, Poland. Their own observations on ultra-long lithium treatment of patients with bipolar disorder are also presented. The review can help psychiatrists to perform a successful lithium prophylaxis in bipolar patients.

## 1. Introduction

Lithium therapy makes the gold standard for the long-term prophylactic (recurrence-preventing) treatment of bipolar disorder (BD), recommended by most guidelines [1,2]. The experiences with long-term lithium therapy, where the cases over 40 or even 50 years of incessant administration have been described, greatly exceed those with other mood stabilizers. Along with the prophylactic efficacy, preventing recurrences of manic and depressive episodes, long-term lithium administration is connected with many adverse but also some beneficial side effects. A comprehensive review of the adverse side effects of lithium was performed by Gitlin in 2016 [3]. The recent meta-analysis on the anti-suicidal effect of lithium treatment was conducted by Smith and Cipriani in 2017 [4], while the antiviral, immunomodulatory, and neuroprotective properties of this drug made the subject of a recent review by Rybakowski [5]. Pharmacokinetic and pharmacodynamic interactions of lithium may be also important for an appropriate lithium administration and its combination with other medications. The recent evaluation of lithium interactions with other drugs was made by Finley in 2016 [6]. This paper is the narrative updated review of lithium-induced side-effects and interactions including the papers on this topic appearing in recent years, including research performed at the Department of Adult Psychiatry, Poznan University of Medical Sciences, Poland. The main focus is put on those making a significant impact on long-term lithium treatment. The prevalence, clinical symptoms, and management strategies for the adverse effects and unfavorable interactions of lithium are described. The aim of the review is to help psychiatrists to perform a successful lithium prophylaxis in bipolar patients. 

## 2. Adverse Side Effects

Adverse side effects of lithium can be divided with respect to the systems or organs, such as renal, gastrointestinal, neurological, thyroid, metabolic, cognitive, dermatological, cardiologic, and sexual. Some occur in the initial period of lithium treatment and a part of them may subside during longer administration. However, several adverse side effects can appear at any stage of lithium maintenance therapy, and a far few of them may have a significant impact on the success of lithium treatment. For managing adverse side effects of lithium, several approaches are recommended. Watchful waiting may be sufficient if tolerance occurs (e.g., nausea). The most frequent procedure is modifying the drug’s administration such as lowering the dose, resulting in the decrease in serum concentration, but also altering the time of its providing or switching to a different lithium formulation. Antidotes for specific side-effects can be employed. In rare cases, the decision can be made of lithium discontinuation or/and change to a different mood stabilizer [3]. 

### 2.1. Renal Side Effects

#### 2.1.1. Polyuria

Lithium reduces renal concentrating ability, resulting in excessive urination (polyuria) with concomitant polydipsia. This effect can appear in the first weeks of lithium therapy, whereas a reduction in urinary specific gravity can in some patients last over years of lithium treatment. In its most extreme manifestation, the polyuria may have a form of diabetes insipidus. The mechanism is due to lithium’s effect on the collecting tubules which generate cyclic adenosine monophosphate in response to antidiuretic hormone. In our study of 78 bipolar patients using lithium for 5–38 (mean 16) years, we found urinary specific gravity of ≤1.005 in 21% of males and 14% of females. A molecular-genetic analysis revealed an association of polyuria with glycogen synthase kinase-3beta (GSK-3β) gene polymorphism [7], which corresponds to the role of this enzyme in the regulation of urinary concentrating ability [8]. 

Polyuria is usually alleviated when the dose of lithium is reduced. If the effects of lithium are very good, a trial of amiloride could be also recommended. In the case of diabetes insipidus, a decision of lithium discontinuation can be made, and following this, the polyuria usually disappears [3].

#### 2.1.2. Nephropathy

The most serious concern in patients on long-term lithium therapy is the possibility of nephropathy in the course of interstitial nephritis. In such conditions, the main changes in laboratory results are increased creatinine concentration and decreased glomerular filtration rate (GFR). The biopsy findings reveal interstitial fibrosis of the cortex and medulla, interstitial changes with benign lymphatic invasion, glomerular fibrosis, and focal nephron atrophy [9]. The renal sonography discloses the presence of microcysts and macrocysts, punctate echogenic foci, and hyperechoic cortex, where the presence of macrocysts is associated with impaired renal function [10]. A recent pharmacogenetic study suggests an association of decreasing GFR during long-term lithium treatment with a polymorphism of the Acid Sensing Ion Channel Neuronal-1 (ACNN1) gene [11].

The International Group for The Study of Lithium Treated Patients (IGSLI) evaluated lithium’s impact on kidneys in patients treated with lithium for 8–48 years (mean 18 years). It was concluded that long-term lithium treatment was associated with a gradual decline of renal functioning by about 30% more than that that due to aging alone. The GFR declined by 0.7%/year of age and 0.9%/year of treatment, both by 19% more among women than men [12]. Recently Rej et al. [13] demonstrated that lithium was associated with a greater risk for impairment of renal function compared to valproate in the population of older adults (mean age 71 years), and the effect was more evident with higher lithium concentrations (>0.7 mmol/L). Swedish investigators studying the cohorts of the LISIE and MONICA projects (more than two thousand patients) concluded that lithium was the commonest cause of moderate-to-severe chronic kidney disease, although comorbidities also played a role. The decline was significantly greater in patients being on more than 10 years of lithium than in all other groups [14]. A meta-analysis by Schoretsanitis et al. [15] showed that one-fourth of patients receiving long-term lithium may develop impaired kidney function, which is two-fold higher compared with the non-lithium population. However, it is thought that generally, the long-term renal outcome of lithium administration in contemporary studies may be better compared with those before the 21st century which may be due to the recommendations for lower prophylactic lithium concentrations [16]. 

In summary, the risk factors for long-term lithium-induced kidney impairment include older age, concomitant comorbidities, higher serum lithium concentrations, and longer duration of lithium treatment [12]. Hayes et al. [17], using a 5-year kidney failure risk equation in a group of over 1600 patients, found predictive factors of high risk of chronic kidney diseases such as younger age at the commencement of lithium, female sex, and lower baseline GFR. The authors point out that individuals at high risk of poor GFR trajectory may be identified before starting lithium treatment. 

In patients in which progressive kidney damage occurs, a decision may be considered to discontinue lithium and replace it with another mood-stabilizing drug. However, such an option should be taken with great caution because other mood-stabilizing drugs, especially in excellent lithium responders, may not be effective, and the further course of the disease may be drug-resistant [3].

In Poznań, a five-year observation of four patients exhibiting very good lithium prophylactic effects was conducted. In three males and one female aged 67–69, with an average lithium treatment duration of 27 ± 9 years, the GFR was less than 50 mL/min/1.73 m^2^. During the follow-up period, in three patients (two males and one female) with an initial GFR of 47–48 mL/min/1.73 m^2^, renal parameters such as GFR, creatinine concentration, and urinary specific gravity did not show any significant changes. Therefore, these patients were advised to continue taking lithium at the current doses and to have an annual check-up of renal condition. In a patient with GFR of 32 mL/min/1.73 s m^2^ whose GFR decreased by 14% and creatinine concentration increased by 10%, the lithium dose was reduced by half, and a periodic consultation with a nephrologist was recommended [18].

Two cases were reported on reintroducing lithium treatment in bipolar patients after renal transplantation due to end-stage renal disease [19,20]. The second case describes the 65-year-old woman, an excellent lithium responder, who after lithium discontinuation was hospitalized more than 40 times within two years and after re-starting lithium therapy was able to achieve remission of symptoms once again [20]. 

The recommendations for monitoring renal status in patients treated longitudinally (ten years or more) with lithium suggest that serum lithium levels measurement as well as those of urea, electrolytes, and creatinine should be performed every 3 to 6 months. A referral to nephrologist’s evaluation should be considered if GFR is <30 mL/min/1.73 m^2^ and a progressive decline in GFR is observed, particularly if the decrease is more than 4/mL/min/1.73 m^2^ [21]. Other cautions should include avoiding combining lithium with medications that can increase nephrotoxic potential (e.g., nonsteroidal anti-inflammatory drugs, angiotensin-converting enzyme inhibitors, some antibiotics as well as cytostatic and immunosuppressive drugs). Acute episodes of renal toxicity should be warded off, and, if possible, lithium should be administered in one daily dose. In the case of exacerbation of chronic kidney disease, the lithium dose must be reduced. Sometimes, even very low lithium doses providing a concentration of 0.2–0.3 mmol/L may be sufficient and a decision to discontinue lithium may not be necessary [21].

### 2.2. Gastrointestinal Side Effects

Gastrointestinal side effects present most commonly as nausea and diarrhea. Nausea occurs in 10–20% of lithium-treated patients being more frequent at the initiation of lithium treatment. The symptom is usually rather tolerated by the patients and gradually subsides during long-term therapy. Nausea may correlate with lithium levels, especially peak levels. Therefore, if it causes a major inconvenience, the patient can be advised to take lithium after meals, to use a multiple daily dose regimen or, if possible, to switch to sustained release preparations. Vomiting is rare in the beginning period of lithium treatment if the lithium level is kept at the therapeutic range. If it occurs, the lithium dose should be reduced. In the first months of treatment, diarrhea may occur in up to 10% of lithium-treated patients. Higher serum lithium levels (e.g., 0.8 mmol/L) may be connected with a predisposition to this symptom and lowering the dose may be recommended. Perhaps in some patients, diarrhea may be associated with sustained-release lithium preparations due to more distal absorption of the drug. In such cases, switching to regular lithium formulation may be helpful. Both vomiting and diarrhea develop in the situation of lithium poisoning [21].

### 2.3. Neurological Side Effects

#### 2.3.1. Tremor

The most common of lithium side effects is tremor, mainly of the hands, reported in about 20–25% of the patients, perhaps more often in men. Usually, it appears in the first few weeks after starting lithium therapy, especially in patients with other predisposing factors such as older age and combined therapy with antipsychotics, antidepressants, and antiepileptic drugs. Some research indicated that hand tremor is related more to brain lithium concentration than serum lithium levels. The tremor is fine, similar to essential tremor, different than Parkinson-type tremor, and most apparent when taking planned actions such as lifting the cup or other precise hand movements. The recommended action to alleviate symptoms of tremors is the reduction of lithium dose. If such treatment is not efficient, the administration of propranolol at doses of 20–80 mg/day is a potentially effective strategy [22]. 

#### 2.3.2. Other

The extrapyramidal symptoms, such as parkinsonism, tardive dyskinesia, or akathisia can also occur during lithium treatment, especially in older age and with combined therapy with neuroleptics. There are also reports on cerebellar symptoms such as nystagmus and gait disturbances, sometimes occurring even with the therapeutic level of lithium. In such cases, if they do not disappear after major reduction of lithium dose, discontinuation of the drug should be considered [22].

#### 2.3.3. Lithium Intoxication

The symptoms of neurotoxicity always appear during lithium poisoning. This can be caused by overdosing, both intentional and accidental, dehydration, infection with fever, and the addition of other drugs. Symptoms appear when lithium concentration reaches 1.2 mmol/L and above. Mild poisoning manifests as weakness, severe hand tremor, mild ataxia, concentration disorders, and diarrhea. Severe poisoning presents with vomiting and neurological disturbances, mainly cerebellar symptoms (tremor, nystagmus, dysarthria, dizziness, ataxia), less often parkinsonian symptoms, choreoathetoid movements, or epileptic seizures. The disturbance of consciousness develops, which is mostly manifested as a stupor. Treatment of lithium poisoning includes intensive somatic care with forced lithium elimination by the infusion of physiological saline. For severe poisoning and high lithium concentration (>2 mmol/L), hemodialysis is very effective for a rapid decrease in lithium concentration [23]. 

In some patients, cognitive disorders and neurological deficits, mostly as symptoms of cerebellar damage (ataxia), may be a sequel of lithium poisoning. For defining such cases, the acronym SILENT (syndrome of irreversible lithium-effectuated neurotoxicity) was coined [24]. Lithium intoxication may also predispose to further kidney damage [9].

### 2.4. Thyroid Side Effects

Adverse lithium effects on the thyroid were observed as early as in the 1960s [25]. Lithium accumulates in the thyroid gland at three- to four-fold higher concentrations than in plasma. Administration of the drug causes a reduced production and release inhibition of thyroid hormones. The most common thyroid side effects associated with long-term lithium treatment are hypothyroidism and goiter. 

The prevalence of hypothyroidism during lithium treatment varies from 6% to 50%, with significantly higher female preponderance. Recently we studied 98 bipolar subjects receiving lithium for mean of 19 years, and 39 subjects, matched for gender and duration of the illness, never receiving lithium. The concentration of the thyroid-stimulating hormone (TSH) was significantly higher in patients receiving lithium. The frequency of hypothyroidism in the course of the illness was similar in both groups (24% vs. 18%) and higher in women than in men (32% vs. 7% and 22% vs. 8%, respectively). This may suggest that bipolar illness itself may predispose to such a condition. In lithium-treated subjects, hypothyroidism usually developed in the first years of lithium therapy, more frequently in subjects with a family history of thyroid dysfunction. All hypothyroid patients were successfully treated with levothyroxine [26].

The prevalence of goiter is estimated at 30–59% of lithium-treated patients. The mechanisms of lithium-induced goiter can include the inhibition of thyroid hormone release resulting in an increase in TSH that leads to the increment of the thyroid volume. Additionally, the activation by lithium of tyrosine kinase and the influence on intracellular signaling by cyclic AMP and Wnt/beta-catenin induces the proliferation of thyrocytes [27]. In our study mentioned above, the thyroid volume was significantly higher in the lithium-treated group than in the lithium-naïve group as were more numerous focal changes >1 cm. The percentage of goiter was significantly higher in the lithium-treated group (40%) than in the lithium-naïve group (18%). In lithium-treated patients, the incidence of goiter was similar in male and female patients (37% vs. 41%). Importantly, the structural changes were not related to the concentrations of thyroid hormones [26]. Additionally, although lithium is supposed to interfere with thyroid immunity, no difference in thyroid autoantibodies was found between the two groups [28].

After starting lithium, it is recommended to monitor thyroid functions every 6 months. The presence of elevated TSH, hypothyroidism, and/or goiter during successful lithium therapy makes by no means evidence to discontinue the drug. All such conditions constitute indications for the thyroxine treatment; the dose can be consulted with an endocrinologist. Whereas, in very rare cases of hyperthyroidism, an endocrinologist’s intervention is a necessity, and possible lithium discontinuation may be considered [23]. 

### 2.5. Metabolic Side Effects

#### 2.5.1. Weight Gain

Weight gain is an inconvenient side effect in lithium-treated patients, and in rare cases, may lead to clinical obesity. In some subjects, especially women, it can significantly impair compliance. The mechanisms of action may include the consumption of high-calorie drinks, insulin-like actions of lithium on carbohydrate and fat metabolism, as well as sodium retention. Bopp et al. [29] showed that in lithium augmentation of antidepressants, weight gain could be associated with leptin gene polymorphism. Whereas, in a recent meta-analysis, it was shown that weight gain on lithium cannot be remarkable and lower than on some active comparators (e.g., valproate) [30]. 

The main preventive and therapeutic strategies include diet, physical activity, and monitoring of thyroid function. Combined therapy with drugs of low weight-gain potential (lamotrigine, aripiprazole) or the addition of topiramate may be considered. Some case reports describe the use of metformin with good results [31].

#### 2.5.2. Calcium and Bone Metabolism

Lithium can increase calcium reabsorption in the kidneys, stimulate parathormone (PTH) secretion, and cause a calcium-phosphate imbalance in the form of primary lithium-induced hyperparathyroidism. A meta-analysis has found that patients undergoing lithium therapy have on average higher concentrations of calcium and PTH [32]. In Poznan, in the study of 90 patients (30 males and 60 females) aged 60 ± 10, and treated with lithium for 16 ± 10 years, hypercalcemia was found in 10% of them (7% males and 12% females). Among the nine patients with hypercalcemia, three were found to have hyperparathyroidism [33]. Lithium-induced hyperparathyroidism can be due to lithium’s action on the calcium-sensing receptor and GSK-3, revealed by unmasking hyperparathyroidism in patients with a subclinical parathyroid adenoma or possibly by initiating multiglandular hyperparathyroidism [34]. Therefore, in people treated with lithium for a long time, it is advisable to monitor serum calcium levels, and in a situation of their increase, also PTH levels. In the case of lithium-induced hyperparathyroidism with significant clinical symptoms, treatment is similar to that of primary hyperparathyroidism.

In contrast to possible disturbances in calcium metabolism, there are some data on the positive effect of lithium on bone mineral density [35]. A recent retrospective cohort study performed in Denmark showed that while bipolar disorder was associated with an increased risk of osteoporosis, the treatment with lithium resulted in a decrease in such risk [36].

### 2.6. Cognitive Side-Effects

Moderate cognitive impairment is perceived by clinicians as associated with lithium treatment. Contrary to that, the results of animal research in various models point to lithium exerting a favorable effect on cognitive functions. Additionally, patients with BD have cognitive problems of various intensity, worsening during manic or depressive episodes, and sometimes also persisting during euthymia. Therefore, this should be adjusted when assessing the lithium effect. It seems that lithium treatment may not negatively affect previously impaired cognitive functions in bipolar patients [37]. Italian authors have carried out neuropsychological tests on lithium-treated bipolar I patients in the state of euthymia, gender- and age-matched patients with the illness receiving other mood-stabilizing drugs, and healthy individuals. It was found that bipolar patients showed deficits in visual memory and executive functions. However, after taking into account the method of treatment, it turned out that these deficits occurred only in patients using mood stabilizers other than lithium [38]. Additionally, in a more recent study, longitudinal monotherapy with lithium resulted in a significant improvement in the global cognitive index score [39].

In Poznan, an attempt was made to correlate the cognitive functions of lithium-treated patients with the quality of lithium prophylactic effect. Lithium non-responders had significantly worse performance on many domains of the Wisconsin Card Sorting Test compared to excellent and partial responders [40]. We also demonstrated that bipolar patients who are excellent lithium responders have cognitive functions comparable to those of matched control subjects [41]. Therefore, an important mechanism of the effect of lithium on cognitive functions could be related to the prevention by lithium of affective recurrences. Previous studies also showed a correlation between neuropsychological deficits, a greater number of affective episodes, and a more severe course of the illness. 

The neurobiological processes connected with the favorable effect of lithium on cognitive functions may include an enhancement of the BDNF system and inhibition of GSK-3β. Interestingly, because of the pathogenic importance of herpes viruses in Alzheimer’s disease [42] and lithium’s antidementia activity (see later), the antiviral effect of lithium against herpes viruses may also be taken into account. It was found that infection with herpes simplex virus type 1 in BD was connected with a decrease in cognitive functioning [43], and the association between cognitive impairment and HSV-1 positivity both in BD and schizophrenia was recently confirmed [44].

From the practical point of view, in cases of suspected cognitive problems in lithium-treated patients the reduction of lithium dose should be considered. The experience shows that using lithium in the lowest useable dose could be a protective factor against possible cognitive problems.

### 2.7. Dermatological Side-Effects

Among the lithium-induced dermatological side-effects, exacerbated acne and psoriasis, as well as de novo occurrence of such conditions, should be mentioned. Psoriasis of moderate to severe intensity may be a contraindication for the introduction of lithium. Dermatological symptoms may be related to the concentration of lithium, and therefore an attempt at dose reduction may be made. If the effects of lithium are good, drugs recommended for these diseases can be used. Only in very severe cases, it may be necessary to replace lithium with another mood stabilizer. Recently, a case of a treatment-resistant female bipolar patient with psoriasis was presented, in whom, after the introduction of lithium, a remission of bipolar illness was achieved as well as a reduction of psoriatic changes. This could be explained by lithium’s immunomodulatory activity [45].

### 2.8. Cardiological Side-Effects

According to a recent review, the most important cardiological side effect of lithium is sinus bradycardia. In persons with genetic predisposition, a clinical presentation of Brugada syndrome may occur. Additionally, electrocardiographic changes (ECG) can appear, such as reversible T wave inversion, sinus node dysfunction, sinoatrial blocks, PR prolongation, QT prolongation/dispersion, and ventricular tachyarrhythmias. The rare cases of ST elevation, myocardial infarction, and heart blocks were also described. The ECG changes are generally dependent on both the duration of lithium treatment and the serum concentration of the drug [46]. Therefore, periodic ECG monitoring in patients during long-term prophylactic lithium treatment is recommended. In severe cases of arrhythmia, if lithium therapy is necessary, cardiological consultation should be considered. 

Recently, studies appeared indicating a possible beneficial effect of lithium on cardiac functions. Taipei’s authors showed that patients who received lithium as the maintenance treatment had significantly lower mean values of left ventricular internal diameters at end-diastole and end-systole, higher mean values of mitral valve E/A ratio, and superior performance of global longitudinal strain than those without lithium treatment. As possible mechanisms, they postulate lithium’s effect on cardiac remodeling involving many signaling pathways as well as its influence on a chemokine, fractalkine [47].

### 2.9. Sexual Side Effects

Sexual side effects are rarely reported by lithium-treated patients, and they also are under-researched. The literature survey from 2015 indicates that lithium may reduce sexual thoughts and desire, worsen erectile function, and diminish sexual satisfaction, and an increased risk of such dysfunctions may appear with concomitant benzodiazepine use. The mechanism behind this should be a reduction by lithium of testosterone concentration and impairing nitric oxide-mediated relaxation of cavernosal tissue [48]. A report from 2014 suggests the prevalence of sexual dysfunction in the group of bipolar patients with mean duration of lithium treatment of 10 years one-third of them [49]. In a recent multicenter cross-sectional study, lithium in monotherapy or in combination with benzodiazepines was found to be related to worse total sexual functioning than anticonvulsants in monotherapy [50]. 

Although sexual dysfunction during lithium treatment appears associated with a lower level of overall functioning and may reduce compliance, limited treatment approaches were suggested. In one study, aspirin 240 daily, was reducing overall sexual dysfunction and improved erectile dysfunction [51]. However, it seems that, especially in diminished arousal, modern phosphodiesterase-5 inhibitors should be considered. 

## 3. Beneficial Effects of Long-Term Lithium Treatment

### 3.1. Anti-Suicidal

Long-term lithium treatment is connected with a significant anti-suicidal effect. It was confirmed by the three meta-analyses performed in the 21st century [4,52,53]. A significantly better effect of lithium than a placebo in decreasing the number of suicides and deaths from any cause both in bipolar disorder and recurrent depression, superior to other mood-stabilizers or antidepressants, was found [4]. Currently, the anti-suicidal effect of lithium in mood disorders is well documented and the largest among other mood-stabilizing drugs. According to German researchers, the anti-suicidal effect of lithium is most prominent after 2 years of lithium administration and is not correlated with the quality of prevention of mood recurrences [54]. As suicides are the most common cause of death of people with mood disorders, this effect of lithium translates into providing significantly lower mortality in patients using it. Therefore, it seems reasonable to consider long-term lithium administration in each patient with a mood disorder having a high suicidal risk.

### 3.2. Antiviral

The antiviral effect of lithium, especially against herpes viruses, has been known since 1980 when British researchers showed that lithium inhibits the replication of the herpes simplex virus (HSV) in an experimental model [55]. In 1991, the “Lithium” journal presented the results of a Polish–American retrospective study of the occurrence of labial herpes in patients receiving lithium for prophylactic purposes. In the Polish sample, nearly half of the patients with recurrent labial herpes experienced a complete elimination of recurrences, and in the remaining—the frequency decreased. A better “anti-herpes” effect was observed in patients with serum lithium concentration over 0.65 mmol/L, and erythrocyte concentration over 0.35 mmol/L. In the American sample, the frequency of recurrences of oral herpes, compared to five years before the treatment, decreased in the group receiving lithium by 73%, while there was no significant difference in the group taking antidepressants [56]. The results obtained may show that long-term lithium treatment may bring about additional therapeutic advantage in bipolar patients with recurrent herpes infections.

In experimental studies, promising results were found for using lithium also against RNA viruses, including coronaviruses [57]. Therefore, interest was kindled in the possible benefits of lithium in COVID-19 patients [58]. Recently, De Picker et al. [59] assessed the relationship between the presence of COVID-19 infection and serum lithium levels in more than 26 thousand patients. The incidence of infection was significantly lower in those with lithium levels maintained within therapeutic limits (0.5–1.0 mmol/L) compared with those with lithium levels <0.5 mmol/L. It may correspond to the relationship between the concentration of lithium and its therapeutic effect on labial herpes [56]. Furthermore, in those with a therapeutic level of lithium, the incidence of infection was significantly lower compared with patients using valproate [59]. Therefore, a possible antiviral effect of lithium in patients with COVID-19 remains a promising issue.

### 3.3. Anti-Dementia

In the 21st century, evidence was also obtained on the neuroprotective and „antidementia” effects of lithium [60]. Lithium treatment was shown to increase the volume of cerebral grey matter, especially that of the prefrontal cortex, anterior cingulate, and hippocampus. Such an effect was not observed for any other mood-stabilizing drug. In the IGSLI study, lithium-treated bipolar patients had a larger volume of the hippocampus than non-lithium patients; such volume in BD subjects was similar to healthy controls [61]. The suggestions were also put forward that lithium could protect against dementia. The association between lithium treatment and a diminished risk of dementia was noticed in a recent 15-year study by Chen et al. [62] including 29,618 patients of whom 548 were exposed to lithium. After controlling for sociodemographic and medical factors it turned out that lithium use was associated with a lower risk of dementia, with a hazard ratio (HR) 0.56, including Alzheimer’s disease (HR = 0.55) and vascular dementia (HR = 0.36). The data from the Danish nationwide register showed that BD patients treated with lithium for a long time had the rate of dementia similar to or lower than in the general population. However, such a rate was increased in patients who received longitudinally anticonvulsant, antidepressant, and antipsychotic drugs [63]. In studies performed in Denmark and the USA, a negative correlation was found between the concentration of lithium in drinking water and the frequency and severity of dementia [64,65]. Finally, possible benefit of using lithium in dementia was found in a recent meta-analysis [66]. 

As bipolar disorder itself increases the risk of dementia, the results obtained may show an additional advantage of the long-term lithium treatment in a form of decreasing such a risk. 

## 4. Lithium Interactions

### 4.1. Pharmacokinetic Interactions

Pharmacokinetic interactions with lithium are connected with the effect of other substances on renal lithium clearance resulting in an increase or decrease in serum lithium concentration. Interactions that increase lithium levels, which may lead to lithium poisoning and cause symptoms of neurotoxicity and organ toxicity, should be of special interest. 

A decrease in lithium clearance can be due to stimulation of proximal tubular sodium reabsorption resulting in concomitant reabsorption of lithium. The drugs reducing lithium clearance are diuretics such as thiazides, amiloride, and spironolactone. Loop diuretics (e.g., furosemide) exert a lesser effect. Such lithium clearance reduction is also caused by angiotensin-converting enzyme inhibitors (ACE-I) and angiotensin receptor blockers (ARB). Drugs that are lowering blood pressure such as ACE-I, beta-blocking agents, or calcium entry blocker verapamil can also decrease lithium clearance by reducing renal perfusion pressure. Other calcium-channel antagonists can cause lithium toxicity; however, the postulated mechanism of action is probably different (an increase in lithium concentration in cells). Furthermore, a decrease in lithium clearance using nonsteroidal anti-inflammatory drugs (NSAIDs) is due to the reabsorption of lithium in the distal neuron being induced by these drugs. Aspirin does not exert such an effect [6]. 

Accelerating lithium elimination (increase in lithium clearance) can be obtained through decreasing lithium reabsorption in the proximal tubule by osmotic diuresis (e.g., mannitol), carbonic anhydrase inhibitor, acetazolamide, and sodium bicarbonate [67]. Some calcium-channel blockers such as nifedipine or isradipine can cause an increase in lithium clearance by producing afferent arteriolar vasodilatation. The same mechanism pertains to xanthines such as aminophylline, theophylline, and caffeine. Patients who abruptly stop excessive drinking of coffee or tea may be at risk of decreasing lithium clearance which may even result in intoxication [6].

Pharmacokinetic interactions between lithium and other drugs are shown in Table 1.

From a practical point of view, the use during long-term lithium treatment of antihypertensive drugs and non-steroidal anti-inflammatory drugs (NSAIDs) is probably the most important. The employment of thiazides, ACE-I, and ARB in combination with lithium is generally not recommended. However, Hommers et al. [68] suggest that a combination of lithium with ACE-I would be possible when sufficient hydration is provided and a combination with hydrochlorothiazide is avoided. Additionally, NSAIDs, which can decrease lithium clearance, are frequently used. Therefore, when the employment of NSAIDs is required for longer time for non-psychiatric reasons, the recommended procedure is to lower the dose of lithium with frequent concentration testing [69].

### 4.2. Pharmacodynamic Interactions

Pharmacodynamic interactions may result from the combination of lithium with other mood stabilizers as well as with antidepressive and antipsychotic drugs. As monotherapy with lithium is prophylactically effective in about 30% of bipolar patients, a combination with other mood-stabilizing drugs is frequently performed for reaching an optimal effect. The combination of lithium with all first-generation (carbamazepine and valproate) and second-generation mood stabilizing drugs (clozapine, olanzapine, quetiapine, aripiprazole, and risperidone) seems pretty safe as for any combinations; there is no significant influence on the level of each drug. Since lithium increases the number of leukocytes, a combination of the drug with carbamazepine or clozapine may prevent leukopenia induced by these substances.

Increased lithium neurotoxicity can be caused by many drugs especially if they are administered in high doses. They include, among others, typical antipsychotics (chlorpromazine and other phenothiazines, haloperidol), tricyclic antidepressants, carbamazepine, and topiramate. The use of lithium with the selective serotonin reuptake inhibitors (SSRIs), e.g., for augmentation of antidepressants in treatment-resistant depression can in rare cases intensify its serotoninergic effect, leading to the symptoms of serotonin syndrome such as tremors, dizziness, agitation, confusion, or diarrhea. 

Other interactions include the enhancement and prolongation of the action of muscle relaxants, both competitive (e.g., pancuronium) and depolarizing (succinylcholine), and in rare cases, it triggers attacks of congenital muscle fatigue. Therefore, a temporary withdrawal of lithium during electroconvulsive therapy (ETC) could be advised. Lithium has been also reported to inhibit the action of amphetamine derivatives [23].

## 5. Ultra-Long Lithium Therapy

In many patients, especially good responders, the treatment with lithium can be a life-long procedure. In 2016, we described five cases (two men and three women, aged 64–79 years) treated with lithium for over 40 years. In four of them, the average lithium concentration was 0.60–0.65 mmol/L, and in one male, 0.7–0.8 mmol/L. The renal status of three women was asymptomatic stage 2 chronic kidney disease; one male patient was diagnosed with stage 3 chronic kidney disease, and the second male had stage 2/3 of kidney impairment, however, without the signs of progress. The thyroid status in four patients was normal. One female patient was diagnosed with Hashimoto’s disease characterized by very high levels of anti-thyroid antibodies and treated with thyroxine. Serum calcium levels were normal or borderline in all patients. Their cognitive functioning was satisfactory, and they continued occupational activity until 55–65 years of age. In all of them, lithium administration had started in the initial years of the illness, which is thought a good prognostic factor for prophylactic efficacy. These cases may show that, in good lithium responders, ultra-long-term treatment with lithium enables good professional and psychosocial functioning, and the possible somatic side effects are manageable [70].

Recently, a case of a 79-year-old female, who reached the 50-year anniversary of lithium treatment was presented. Lithium treatment was started after the third depressive episode which appeared within four years. The treatment resulted in a complete prevention of depressive or manic episodes, enabling the patient to lead a satisfying personal and professional life. The patient worked successfully as an ophthalmologist until 65 years of age. The kidney status in 2012 was asymptomatic stage 2 chronic kidney disease, and the thyroid function in 2013 was within the normal range as was serum calcium level. In this person, apart from maintaining a euthymic state, long-term treatment with lithium also exerted a favorable effect on general health preventing viral and other respiratory infections as well as producing a disappearance of recurrent herpes. Among the factors that facilitated a good response to lithium, a hyperthymic personality and early start of lithium treatment can be listed. The patient may be regarded as the epitome of an excellent lithium responder [71]. 

## 6. Conclusions

Long-term lithium therapy is the maintenance treatment of choice in bipolar mood disorder which can be successfully performed for 40 years or more. Promising results were also obtained for the long-term prophylaxis of recurrent depression [72]. As shown in this review, such treatment may be connected with several adverse side effects and interactions but also with some beneficial events. The adverse side effects and interactions, with proper knowledge, can be successfully managed. Such an expertise could help the psychiatrist to perform successfully lithium prophylaxis for mood disorders.

Unfortunately, despite clinical efficacy surpassing other mood-stabilizing drugs, manageable side effects and interactions, as well as advantageous properties, that of anti-suicidal would be the most important; lithium currently is greatly under-prescribed. Therefore, its therapeutic potential in mood disorders is not fully utilized. The claims for allowing a larger number of mood disorder patients to became beneficiaries of lithium use have been voiced, also by the authors of this paper [73,74]. It may be hoped that this review could also serve such a purpose. 

## Figures and Tables

**Table 1 pharmaceuticals-16-00074-t001:** Pharmacokinetic interactions between lithium and other drugs (the effect via renal lithium clearance).

Increase in Lithium Concentration and Toxicity (Decrease in Lithium Clearance)	Decrease in Lithium Concentration (Increase in Lithium Clearance)
Diuretics: thiazides, amiloride, spironolactone Angiotensin converting enzyme inhibitors (ACE-I): captopril, enalapril, lisinopril, ramipril et al. Angiotensin receptor blockers (ARB): losartan, valsartan Non-steroidal anti-inflammatory drugs (NSAID): 1. indomethacin 2. diclofenac 3. piroxicam 4. ibuprofen	Xanthines: aminophylline, caffeine, theophylline Calcium channel blockers: nifedipine, isradipine Carbonic anhydrase inhibitors: acetazolamide Osmotic diuretics: glucose, mannitol Sodium bicarbonate

## Data Availability

Not applicable.

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
