# Peer review of "Long-Term Lithium Therapy: Side Effects and Interactions"

_pharmaceuticals, 2023, doi:10.3390/ph16010074_

Round 1

Reviewer 1 Report

Thank you for giving me the opportunity to read and comment a report “Long-term lithium therapy: side effects and interactions”, by Ferensztajn-Rochowiak E and Rybalowski J.K.

In the reviewed manuscript, the impact of modulating the impact of long-term lithium treatment is reported.

This paper is well written, correctly structured with a suitable research concept and definitely it is of relevance to readers of the journal. The references used are relevant although more than 50% are dated more than 5 years ago. However, some suggested minor changes are included in the comments given below.

·    Both in line 18 of the abstract and line 38 of the introduction, the authors mention the Poznan center. More information, such as the type of center and indicating that Poznan is a Polish city, would be helpful to the reader.

·         As a recommendation to the authors, it would be interesting to finish the abstract highlighting a main conclusion of the review.

·  In lines 23-24 the authors indicate that lithium is the long-term prophylactic treatment of bipolar disorder, according to guidelines. However, reference 2 provided by the authors is not a guideline, it is a systematic review of randomized trials and meta-analysis.

·      It would be desirable for the authors to provide more detail on the main aim of the study.

·        Acronyms are usually included in parentheses after the corresponding definition, as for example the authors do in lines 75-76. However, in lines 81-82, the authors include the acronym first and then the definition. The same style should be followed throughout the manuscript.

·       There are some paragraphs in the manuscript that are not associated with a reference, e.g.:

o   Lines 68-71.

o   Lines 109-113.

o   Lines 134-141.

o   Liner 194.

o   Lines 225-230.

o   …..

   Please review the manuscript and ensure that all text is properly referenced.

·    According to the journal's instructions, the "author contributions" section is incomplete.

Author Response

  1. Poznan center was replaced by Department of Adult Psychiatry, Poznan University of Medical Sciences, Poland.
  2. The last sentence of the abstract makes a conclusion.
  3. Reference 2 was replaced by a guideline paper.
  4. The main aim of the study was given (lines 44-45).
  5. The acronym was put after the definition in line 86.
  6. The references were provided for the paragraphs mentioned by the reviewer.
  7. Author contributions were expanded.

Reviewer 2 Report

This manuscript systematically reviewed the adverse and beneficial side effects of lithium prescribed for treating bipolar disorder. In addition, potential interactions with concomitantly administered drugs are described.

The authors reviewed these issues well based on existing documents including the most recent published papers. However, it may be better for the readers to put one or two tables summarizing current findings on these issues—especially drug-drug interactions.  

In addition, typographical mistakes should be corrected (for example, p 10:line 490: ‘nad’ should be changed to ‘and’)

Author Response

  1. The table on pharmacokinetic interaction between lithium and other drugs was provided
  2. Typographical mistakes were corrected.